# Hedgehogs and Squirrels as Hosts of Zoonotic *Bartonella* Species

**DOI:** 10.3390/pathogens10060686

**Published:** 2021-06-01

**Authors:** Karolina Majerová, Ricardo Gutiérrez, Manoj Fonville, Václav Hönig, Petr Papežík, Lada Hofmannová, Paulina Maria Lesiczka, Yaarit Nachum-Biala, Daniel Růžek, Hein Sprong, Shimon Harrus, David Modrý, Jan Votýpka

**Affiliations:** 1Department of Parasitology, Faculty of Science, Charles University, 12800 Prague, Czech Republic; jan.votypka@natur.cuni.cz; 2Institute of Parasitology, Biology Center, Czech Academy of Sciences, 37005 Ceske Budejovice, Czech Republic; honig@paru.cas.cz (V.H.); ruzekd@paru.cas.cz (D.R.); modryd@vfu.cz (D.M.); 3Military Health Institute, Military Medical Agency, 16200 Prague, Czech Republic; 4Koret School of Veterinary Medicine, The Hebrew University of Jerusalem, Rehovot 76100, Israel; ricardo.gutierrez@mail.huji.ac.il (R.G.); yaarit.biala@mail.huji.ac.il (Y.N.-B.); shimon.harrus@mail.huji.ac.il (S.H.); 5Center for Infectious Disease Control, National Institute for Public Health and the Environment, 3721 Bilthoven, The Netherlands; manoj.fonville@rivm.nl (M.F.); hein.sprong@rivm.nl (H.S.); 6Department of Infectious Diseases and Preventive Medicine, Veterinary Research Institute, 62100 Brno, Czech Republic; 7Department of Pathology and Parasitology, Faculty of Veterinary Medicine, University of Veterinary Sciences, 61242 Brno, Czech Republic; Reptimania@email.cz (P.P.); lada.hurkova@seznam.cz (L.H.); lesiczkapaulina@gmail.com (P.M.L.); 8Central European Institute of Technology, University of Veterinary Sciences, 61242 Brno, Czech Republic; 9Department of Botany and Zoology, Faculty of Science, Masaryk University, 61137 Brno, Czech Republic

**Keywords:** *Bartonella grahamii*, *B. melophagi*, *B. rochalimae*, *B. washoensis*, ‘*Candidatus* B. rudakovii’, hedgehogs, squirrels, multiple PCR, vector-borne diseases, zoonoses

## Abstract

Free-living animals frequently play a key role in the circulation of various zoonotic vector-borne pathogens. Bacteria of the genus *Bartonella* are transmitted by blood-feeding arthropods and infect a large range of mammals. Although only several species have been identified as causative agents of human disease, it has been proposed that any *Bartonella* species found in animals may be capable of infecting humans. Within a wide-ranging survey in various geographical regions of the Czech Republic, cadavers of accidentally killed synurbic mammalian species, namely Eurasian red squirrel (*Sciurus vulgaris*), European hedgehog (*Erinaceus europaeus*) and Northern white-breasted hedgehog (*Erinaceus roumanicus*), were sampled and tested for *Bartonella* presence using multiple PCR reaction approach targeting several DNA loci. We demonstrate that cadavers constitute an available and highly useful source of biological material for pathogen screening. High infection rates of *Bartonella* spp., ranging from 24% to 76%, were confirmed for all three tested mammalian species, and spleen, ear, lung and liver tissues were demonstrated as the most suitable for *Bartonella* DNA detection. The wide spectrum of *Bartonella* spp. that were identified includes three species with previously validated zoonotic potential, *B. grahamii*, *B. melophagi* and *B. washoensis*, accompanied by ‘*Candidatus* B. rudakovii’ and two putative novel species, *Bartonella* sp. ERIN and *Bartonella* sp. SCIER.

## 1. Introduction

Bartonellae are vector-borne, facultative intracellular, fastidious, gram-negative alpha-2-proteobacteria with significant zoonotic potential. The genus *Bartonella* represents an ecologically successful group of bacteria distributed worldwide that has been found in a wide spectrum of blood-feeding arthropods and vertebrates, including humans [1,2,3] (Appendix A).

Currently, there are 53 proposed *Bartonella* species and seven subspecies with a taxonomic name recognized by the National Center for Biotechnology Information database [4]. Of these, only 37 species and three subspecies have been validly published under the rules of the International Code of Nomenclature of Bacteria (Bacteriological Code) [5]. New species and subspecies from all over the world are continuously being recognized. As candidate species from a wide range of animal reservoirs have been described but not yet assigned, the number of described taxa is steadily growing [6].

In mammals, *Bartonella* infection is characterized by a long-term persistence in erythrocytes and endothelial cells. This protects the bacteria from rapid clearance by the host’s immune system [7,8], making mammals suitable reservoir hosts. Long-lasting bacteremia facilitates the transmission of bartonellae by various blood-feeding arthropod vectors between different reservoirs as well as non-reservoir hosts [9,10]. It is anticipated that numerous arthropods are involved in the transmission, such as sand flies, lice, hippoboscid flies, bed bugs, ticks and above all fleas [3,11]. Moreover, some *Bartonella* species remain infectious in arthropod feces and can infect superficial wounds, e.g., through scratching [3].

Anthroponotic *Bartonella bacilliformis* and *B. quintana* and zoonotic *B. henselae* are the most well-known causative agents of human bartonellosis and cause the vast majority of human disease cases attributed to bartonellae [6]. Nevertheless, in the last two decades, human diseases linked to other *Bartonella* species (*B. alsatica*, *B. ancashensis*, *B. clarridgeiae*, *B. doshiae*, *B. elizabethae*, *B. grahamii*, *B. koehlerae*, *B. kosoyi*, *B. mayotimonensis*, *B. melophagi*, *B. rattimassiliensis*, *B. rochalimae*, *B. schoenbuchensis*, *B. tamiae*, *B. tribocorum*, *B. vinsonii* and *B. washoensis*) have been described, particularly in immunocompromised patients, with mild to serious symptoms (details provided in Appendix A). Currently, bartonellosis is considered an emerging zoonotic disease (e.g., [12,13]). It has even been hypothesized that any of the *Bartonella* species is potentially capable of infecting humans [6,11].

*Bartonella* species have been detected in (or isolated from) a wide range of mammalian hosts, including humans, carnivores, rodents, bats, ungulates, kangaroos and also marine animals such as sea otters and beluga whales [3,6,10,14,15]. Sea turtles are the only nonmammalian vertebrates in which *Bartonella* infection has been detected [16]. Nevertheless, small mammals (especially rodents) are considered one of the most common mammalian reservoirs for *Bartonella* spp. worldwide. The remarkably high infection rate, typically long-lasting bacteremia and vertical transmission observed in some rodents, along with the high level of diversity and frequently mixed infection of different *Bartonella* species and genotypes, strongly indicate that rodents play a key role as reservoir hosts [2,17,18,19,20].

Synurbic animals, which are frequently infested by ticks and fleas, represent a source of pathogens transmissible to humans. They also constitute a valuable source of biological material for vector-borne pathogen monitoring [21,22,23]. In our surveillance, cadavers of hedgehogs and squirrels found in several different areas of the Czech Republic (mostly urban and suburban) were tested for the presence of *Bartonella* DNA. Hedgehogs (*Erinaceus europaeus* and *Erinaceus roumanicus*) and squirrels (*Sciurus vulgaris*) tend towards synurbization, and they are often even more abundant in urban (and rural) areas than in natural habitats such as forests (e.g., [24,25]). These mammalian species are found close to humans (in backyards, urban green spaces, etc.) and are well-liked by urban populations (e.g., people feed them, or in the case of hedgehogs, take care of them during overwintering). As such, they constitute a potential source of zoonotic infections, including those transmitted by arthropods (e.g., [21,26,27]). Various species of the Sciuridae family have been described as hosts of *Bartonella* spp. in countries all over the world (e.g., [28]). In the Eurasian red squirrel, *B. washoensis* and *B. grahamii* have been detected to date [29,30,31,32]. Furthermore, *B. washoensis* has been detected in squirrel ectoparasites, *Ixodes ricinus* ticks and *Ceratophyllus* (*Monopsyllus*) *sciurorum* fleas [32]. There is only limited information about *Bartonella* infection in hedgehogs. In Algeria, *B. elizabethae* and *B. tribocorum* were detected in *Atelerix algirus* [33] and, furthermore, *B. elizabethae* and *B. clarridgeiae* were detected in hedgehog fleas (*Archeopsylla erinacei*) collected from these hedgehogs [34]. In Israel, a single unique sequence was obtained from *Erinaceus concolor*, clustering with *B. clarridgeiae* and *B. rochalimae*, but a more precise determination was not possible [35]. Two sequences (MF372764 and MF372765) closely related to *B. taylorii* were detected in *E. roumanicus* from Hungary [21], and *B. henselae* was identified in an *A. erinacei* flea collected from the same hedgehog species [36].

The main aim of this study was to verify the role of hedgehogs and squirrels as bartonellae hosts and to confirm the suitability of cadavers for *Bartonella* detection. Through multiple PCR targeting several DNA loci, the most suitable tissues for *Bartonella* detection and species identification were determined. The distribution, specificity and zoonotic potential of the detected *Bartonella* species are discussed considering various hosts worldwide.

## 2. Results

Dissection of 55 Eurasian red squirrels, 83 European hedgehogs and 42 Northern white-breasted hedgehogs resulted in a total of 1429 tissue samples (Appendix A).

### 2.1. Tissue Detection Efficiency

To investigate detection efficiency, various tissues from each cadaver were screened for the presence of *Bartonella* DNA using real-time PCR. Ear/skin, muscle, lung, liver, spleen, urinary bladder, kidney and brain tissue and blood were screened from most of the animals (Figure 1), but in some cases, not all tissues were available (Appendix A).

### 2.2. Bartonella Infection Rates and Species Identification

The tissue with the lowest Cp value of each real-time PCR positive individual (with the exception of seven animals, where spleen was used preferentially, Appendix A) was subsequently tested using a set of six different conventional PCR reactions targeting various loci: citrate synthase (*gltA*), the beta subunit of RNA polymerase (*rpoB*), and the cell division protein (*ftsZ*) encoding genes and the 16S–23S rRNA intergenic spacer (ITS). The ITS and *gltA*-1 were the most sensitive assays (positive PCR product in 61% and 60% of samples that were previously real-time PCR positive, respectively) and species identification by *gltA*-1, *gltA*-2 and *rpoB* assays was the most successful (Figure 2 and Figure 3). However, each of the chosen PCR protocols revealed positivity or allowed particular *Bartonella* species identification for at least one host individual missed by all other protocols, e.g., *B. washoensis*-specific PCR revealed four extra infections in squirrels (Figure 3).

In the tested set of real-time PCR positive samples (Figure 3), different conventional PCR reactions revealed the DNA of six *Bartonella* genospecies, including several mixed infections (Figure 3, Figure 4 and Figure 5). For two squirrels, five European hedgehogs and four Northern white-breasted hedgehogs considered to be *Bartonella*-positive by real-time PCR, amplification was successful in at least one of the conventional PCR reactions. However, the low sequence quality (identified as *Bartonella* spp. only) did not allow for accurate species identification (presented in Figure 3 and Figure 4 as “not identified”). In nine cases, the real-time PCR positive samples, which had not been amplified in any of the subsequent conventional PCRs or where the sequencing revealed misamplification of the host DNA, were considered negative for all analyses (Figure 3).

The highest *Bartonella* spp. infection rate was observed in *S. vulgaris* (76%, 42/55 individuals), followed by *E. roumanicus* (43%, 18/42) and *E. europaeus* (24%, 20/83) (Figure 4).

### 2.3. Identified Bartonella Species

A total of six *Bartonella* species were identified by DNA sequencing. Three well-established species, *B. washoensis*, *B. grahamii* and *B. melophagi*, were accompanied by ‘*Candidatus* B. rudakovii’ and two putative novel species, provisionally designated here as *Bartonella* sp. ERIN and *Bartonella* sp. SCIER. No obvious pattern in their geographical distribution was observed in this study (Appendix A).

*B. washoensis* was the only species for which sequences were obtained for all PCR targeted genes and ITS (Figure 3). All of the sequences (represented by CZ-12, CZ-22 and CZ-146 (*gltA*), CZ-259 and CZ-262 (*rpoB*), CZ-23 and CZ-32 (*ftsZ*) and CZ-175 (ITS); for detailed information about all representative sequences see Appendix A) achieved >99% similarity to the *B. washoensis* GenBank sequences. Phylogenetic analysis of the *gltA* sequences revealed two closely related genotypes represented by CZ-12 and CZ-146 within the *B. washoensis* clade (Figure 5). While the CZ-12 and CZ-146 sequences were obtained by species-specific PCR, the CZ-22 was a product of *gltA*-1 PCR targeting a diverse part of the citrate synthase gene. Thus, we were not able to confirm whether the CZ-22 belongs to one of the genotypes represented by CZ-12 and CZ-146 or not. *B. washoensis* was detected by different PCR assays in most of the *Bartonella*-positive squirrels (approximately 67%) and also in two European hedgehog individuals (Figure 3, Figure 4, Figure 5 and Appendix A).

*B. melophagi* was detected in the sole European hedgehog (approximately 1%) by amplifying and sequencing the *gltA* target. The corresponding sequence CZ-87 showed 100% similarity to *B. melophagi* sequences available in the GenBank (MT635399 and AY724768). ‘*Candidatus* B. rudakovii’ (EF682090) was the closest match (97.4%–99.1% similarity) for several sequences obtained exclusively from squirrels, represented by CZ-30, CZ-51 and CZ-52, obtained through *gltA* and *rpoB* targeted PCR protocols (Figure 3, Figure 4, Figure 5 and Appendix A). *Bartonella grahamii* was detected in two squirrels; the obtained sequences (represented by CZ-124) showed 97.7% similarity to *B. grahamii* GenBank sequences and clustered to the *B. grahamii* clade in the phylogenetic analyses (Figure 5).

The two putative novel *Bartonella* species, *Bartonella* sp. ERIN and *Bartonella* sp. SCIER, are represented by *gltA* sequences CZ-47 and CZ-41, respectively (Figure 5). For *Bartonella* sp. SCIER, *rpoB* sequences were also obtained representing three slightly different genotypes (CZ-3, CZ-123 and CZ-232; Appendix A). *Bartonella* sp. SCIER was detected in six squirrels, one European hedgehog and six Northern white-breasted hedgehogs. The sequences obtained by *gltA*-1 PCR assay (represented by CZ-41) showed 98.5% similarity to the *B. grahamii* GenBank sequences (Appendix A). However, phylogenetic analyses indicated rather an unstable position concerning the *B. grahamii* sequences (Figure 5; data not shown). The three *rpoB* sequences also had the closest match to *B. grahamii* (96.1%–96.9% similarity, Appendix A). *Bartonella* sp. ERIN was detected in both hedgehog species as the most prevalent *Bartonella* species (55% of all positive *E. europaeus* and 44% of *E. roumanicus*) and formed a well-supported clade along with *B. clarridgeiae* (95.6% similarity), *B. rochalimae* and ‘*C.* B. rudakovii’ (Figure 5).

## 3. Discussion

In parallel with ongoing global changes, novel vector-borne diseases are emerging, and knowledge about the distribution of reservoirs, vectors and pathogens is becoming a key priority for infectious disease control. This trend is further supported by growing interest and new diagnostic approaches. The urban environment is characterized by a high abundance of relatively few vertebrate species that co-occur in communities that do not exist naturally. Thus, free-ranging vertebrates living in urban ecosystems can serve as reservoirs for zoonotic pathogens. This likely has important yet unknown consequences for the transmission dynamics of vector-borne pathogens, particularly for those which can cause persistent infections and have a relatively broad host range. This holds true for *Bartonella* spp., which is transmitted by various arthropod species and/or via animal bites and scratches and which displays high zoonotic potential (Appendix A). In addition, the spectrum of known vertebrate hosts is growing, both for many well-established *Bartonella* species and for newly recognized ones.

The species diversity of *Bartonella* spp. detected in the three synurbic target hosts (*E. europaeus*, *E. roumanicus*, *S. vulgaris*) generally aligns with previously published results [29,30,31,32,34,35]. The suitability of these mammalian species’ carcasses for *Bartonella* DNA screening was demonstrated previously by Szekeres et al. [21], von Loewenich et al. [31] and Lipatova et al. [32]. However, in comparison to these studies, we noticed higher infection rates (76%, 24% and 42% for Eurasian red squirrel, European hedgehogs and Northern white-breasted hedgehogs, respectively), most likely owing to the diagnostic approach combining real-time PCR and multiple conventional PCRs. There is no obvious explanation for the observed differences in the infection rates (and *Bartonella* spp. diversity) between the two hedgehog species, as none of the analyzed parameters (animal age, geographical origin, grade of autolysis) differed significantly (Appendix A; data not shown). 7% (3/45) of squirrels, 17% (4/24) of European hedgehogs and 10% (2/20) of Northern white-breasted hedgehogs, which had previously been positive in real-time PCR, showed negative results for all six conventional PCRs. This was probably due to the higher sensitivity of real-time PCR in comparison to conventional PCRs, in which samples with low bacteremia may be below the detection limit. Discrepancies in the results from these two methods have been observed for different pathogens and the same samples previously [22,23] and for *Bartonella* spp. in other studies (e.g., [37]). Lipatova et al. [32] also compared the detection efficiency of a particular squirrel’s tissue, showing that lung (47%), bladder (44%) and spleen (39%) samples tested positive for *Bartonella* more frequently than liver, kidney and ear samples (19%, 14% and 11%, respectively). Our results differed slightly, with the highest detection efficiency for squirrels was found for spleen tissue (85%), followed by liver (71%), lung (70%) and ear (68%) tissues (Appendix A). However, a different assay using only ITS PCR was used in the compared study. Moreover, tissue tropism is not expected to occur in the case of bartonellae, as it is considered a blood and endothelial cell parasite. Nevertheless, within our survey, blood and ear tissues from 29 European hedgehogs and 28 Northern white-breasted live-trapped hedgehogs were used as a control group for pathogen detection (for methodology and ethical statement, see [23]). Surprisingly, only four European hedgehogs (i.e., one blood sample and three ear tissue samples) were *Bartonella*-positive according to the real-time PCR results, with the only single sample confirmed by a set of conventional PCRs (data not shown). The skin, muscle, blood, liver and brain tissues from the total of 85 blackbird (*Turdus merula*) and 11 song thrush (*Turdus philomelos*) carcasses [22,23] were also tested for *Bartonella* DNA presence with the same approach used for the mammals in this study. None of these samples were positive, supporting the hypothesis that only mammals (and some reptiles) are hosts for bartonellae, but not birds.

*Bartonella* spp. identification solely through molecular methods is considered to be a challenging task. It has also been shown that sensitivity varies considerably among different PCR assays, and the primers used may favor detection of specific species in mixed infections [38]. Although we tried to avoid this limitation by using PCRs targeting four different loci, our results may be biased due to the selection of assays more suitable for *Bartonella* species occurring in squirrels (i.e., *B. washoensis*). In the comparison of five *Bartonella* spp. conventional PCR protocols used in this study, the *gltA*-1 and ITS protocols showed the highest sensitivity (approximately 60%). On the contrary, the *ftsZ* PCR assay successfully amplified only approximately 6% of the positive samples (Figure 2). Nevertheless, all of the chosen assays enabled us to identify the infection by an extra *Bartonella* species for at least one sample/specimen, including the *B. washoensis*-specific PCR protocol. The combination of *gltA*-1 and *gltA*-2 assays revealed all six detected species (four and three, respectively), and similarly, three species were detected by the *rpoB* assay. We conclude that the multiple PCR reactions approach revealed higher *Bartonella* species diversity, which would be missed if only a sole PCR assay were used (Figure 3).

Thus far, human infections by *B. washoensis* have been recorded in three cases: two in North America [39,40], where Californian ground squirrel (*Otospermophilus beecheyi*) seems to be a reservoir [41], and a single human European case recently described in Germany [31]. The relatively high prevalence of *B. washoensis* in squirrels in the Netherlands, United Kingdom and Lithuania suggested that *S. vulgaris* may be a vertebrate reservoir of zoonotic *B. washoensis* in Europe [29,31,32]. Our findings of a high proportion of *B. washoensis* bacteria in squirrels strongly support this hypothesis. However, detection of *B. washoensis* in other mammalian species (various members of the Sciuridae family [28] and dogs [42]) indicate rather lower host specificity. Our results from the European hedgehog represent the first finding of *B. washoensis* in insectivores and thus extend the recognized host spectrum.

*B. melophagi* has been frequently found in sheep [43,44] and sheep keds, *Melophagus ovinus* [44,45,46,47,48,49], and some authors hypothesize that this *Bartonella* species is highly host- and vector-specific [3,44,50]. Nevertheless, *B. melophagi* infections may cause human disease [50]. They were also recently detected in white-tailed deer *Odocoileus virginianus* [51], forest flat fly *Hippobosca equina* [49] and questing ticks of two species, *Haemaphysalis qinghaiensis* and *Dermacentor everestianus* [18]. Our findings of *B. melophagi* in a European hedgehog contribute to the discussion about host specificity. 

Two of the detected *Bartonella* species belong to the *B. rochalimae* clade (Figure 5 and Appendix A). One is closely related to ‘*C.* B. rudakovii’ while the second shows the highest similarity to *B. clarridgeiae*. There is almost no information about ‘*Candidatus* B. rudakovii’ in the literature, apart from the GenBank database (EF682090) data concerning detection of this candidate species in bank voles (*Clethrionomys glareolus*) in Western Siberia. According to the sequence similarity, this species was also detected in *C. glareolus* in Lithuania (*rpoB* sequence MH547308; [37]). In our analysis (Figure 5 and Appendix A), the sequence of *Bartonella* sp. AR-15 (FN645480) detected from *Tamiasciurus* sp. [52] also shows close similarity to ‘*Candidatus* B. rudakovii’ and likely represents the same species (or its variant). All these sequences were the closest GenBank matches for several sequences obtained from eleven squirrels in this study (represented by CZ-30 and CZ-51 (*gltA*) and CZ-52 (*rpoB*)), and thus could be considered as variant(s) of ‘*Candidatus* B. rudakovii’. Since we were able to detect this *Candidatus* species by three different PCR protocols (*gltA*-1, *gltA*-2, *rpoB*) in one individual (squirrel no. 51), we suppose all of the sequences (CZ-30, 51 and 52) may constitute a single variant or even genotype. However, mixed infections with several *Bartonella* species or strains in one animal are common, and there is always the possibility of getting amplicons from the same sample that actually belong to different *Bartonella* species or genotypes coexisting in the same individual. However, all these sequences also show high similarity with *B. rochalimae*, a well-recognized zoonotic *Bartonella* species reported as a causative agent of human disease [53,54]. It has been suggested that wild carnivores serve as major reservoirs for *B. rochalimae* (e.g., [8,55]) since this species has been detected mostly in canids, raccoons and skunks (e.g., [35,55,56,57,58,59,60,61,62]). On the other hand, multiple recent studies have shown that other small mammals, including rodents, can also harbor *B. rochalimae* [35,63,64,65,66,67,68,69,70]. Therefore, our findings in Eurasian red squirrels are not surprising and indicate that small mammals can play an important role in the circulation of species from the *B. rochalimae* clade.

The closest GenBank match (95.59%) for *gltA* sequences represented by CZ-47 and designated here as *Bartonella* sp. ERIN was *B. clarridgeiae*, a well-known pathogen of humans (Appendix A), cats (e.g., [71]), dogs (e.g., [72]), and rodents [30,73,74]. According to the 96% *Bartonella* species threshold criterion for *gltA* sequences proposed by La Scola et al. [75], *Bartonella* sp. ERIN might be considered as a new species. However, this criterium also includes <95.4% identity for the *rpoB* gene. Furthermore, bacterium speciation requires bacteria isolation and full genetic and biochemical characterization, which we are not able to provide [38].

*B. grahamii* is considered to be a rodent-borne zoonotic bacterium found in a multitude of rodent genera (*Apodemus*, *Arvicola*, *Clethrionomys*, *Cricetulus*, *Dryomus*, *Micromys*, *Microtus*, *Mus*, *Pteromys*, *Rattus* and *Sciurus*) worldwide (e.g., [37,76,77,78,79]). Our findings in two squirrels confirmed previous reports. *B. grahamii* was also the closest match for *gltA* sequences represented by CZ-41 (98.5% similarity) and several *rpoB* sequences (represented by CZ-3, CZ-123 and CZ-232; 96.1%–96.9% similarity) obtained from all three targeted mammalian species and designated here as *Bartonella* sp. SCIER. According to the BLASTn analysis only, these sequences could be considered variants of *B. grahamii*. On the contrary, phylogenetic analyses indicated *Bartonella* sp. SCIER as a distinct species (Figure 5 and Appendix A). However, a culture isolation method is required for more accurate identification of the most likely new *Bartonella* sp. SCIER species.

Urban and peri-urban ecosystems represent dynamically evolving environments with growing numbers of synurban vertebrates. We demonstrated that cadavers of synurban mammals represent a valuable source of biological material for vector-borne pathogen screening. High infection rates of *Bartonella* spp., confirmed for all three tested mammalian species, and the discovery of two presumably new *Bartonella* species call for a more robust system of involvement of accidentally killed urban vertebrates in pathogen surveillance.

## 4. Materials and Methods

A total of 180 cadavers of the European hedgehog (*Erinaceus europaeus*), the Northern white-breasted hedgehog (*Erinaceus roumanicus*) and the Eurasian red squirrel (*Sciurus vulgaris*) were collected in various geographical regions of the Czech Republic for a project addressing the role of these species in the circulation of vector-borne pathogens and their possible use as sentinel hosts. All of these animals were found dead, mostly as roadkill, killed by other animals (above all by dogs and cats) or had died in animal rescue centers. The cadavers were kept at −20 °C until processing. Cadaver collection, dissection, tissue sample preparation and DNA extraction were performed as described previously [22,23]. 

All PCR reactions were set up in separate rooms with all required precautions (separated supplies, equipment, and personal safety items, pre- and post-amplification activities). Real-time PCR targeting a fragment of the *ssrA* (transfer-messenger RNA) gene [80] with forward and reverse primers (*ssrA-F* 5′-GCT ATG GTA ATA AAT GGA CAA TGA AAT AA-3′; *ssrA-R* 5′-GCT TCT GTT GCC AGG TG-3′) and probe (5′-ATTO520-ACC CCG CTT AAA CCT GCG ACG-BHQ1-3′) in 10 μM concentration was used to detect *Bartonella* spp. in all tissue samples. For real-time PCR, an iQ Multiplex Powermix PCR reagent kit containing iTaq DNA polymerase (Bio-Rad Laboratories, Hercules, CA, USA) was used. Real-time PCR was performed in a LightCycler 480 Real-Time PCR System (Roche, Basel, Switzerland) under the following conditions: an initial activation of the iTaq DNA polymerase at 95 °C for 5 min, 60 cycles of a 5 s denaturation at 94 °C, followed by a 35 s annealing-extension step at 60 °C (single point measurement at 60 °C), and a cooling cycle of 37 °C for 20 s. The analysis was performed using second derivative calculations for Cp (crossing point) values. Curves were inspected visually in LightCycler 480 software. A negative control (no template) and positive control (*B. henselae* DNA) were added to each set of samples.

Hedgehogs and squirrels that tested positive for *Bartonella*-DNA by real-time PCR (at least in one tested tissue sample) were subsequently analyzed by six different conventional PCR assays targeting citrate synthase (*gltA*), the beta subunit of RNA polymerase (*rpoB*), and the cell division protein (*ftsZ*) encoding genes and the 16S–23S rRNA intergenic spacer (ITS) (Table 1). For these analyses, the tissue sample with the lowest Cp value (most frequently spleen and ear tissues, Appendix A) per animal was used. The cycling conditions for the individual primer pairs are described in Appendix A. A sample/animal was considered to be positive when at least one of the conventional PCRs and subsequent sequencing (including low-quality sequences that could be identified as *Bartonella* sp., but not to the species level, because of the fragment size or the number of ambiguous nucleotides) confirmed the real-time PCR positive results.

The EmeraldAmp GT PCR Master Mix (TaKaRa Bio Inc., Shiga, Japan) was used for all conventional PCRs according to the manufacturer’s instructions. A total volume of 20 μL was prepared for each reaction containing 10 μL of the master fco, 10 pmol of each primer, 2 μL of template DNA or 1 μL of PCR product from the first PCR in the case of nested PCR (*gltA*-1, *rpoB*, ITS) or second PCR (*ftsZ*), filled with PCR-grade water. The PCR products were visualized using 1.5% agarose gel electrophoresis under UV light, cleaned with ExoSAP-IT™ PCR Product Cleanup Reagent (Applied Biosystems™, Waltham, MA, USA), and direct Sanger sequencing was provided by SEQme (Dobris, Czech Republic), BaseClear (Leiden, Netherlands), or Hy Laboratories Ltd. (Rehovot, Israel). The sequences were checked manually using Geneious software (version 10.0.6) and compared to sequences from the NCBI GenBank database for *Bartonella* species identification. All the partial *gltA* gene sequences of related recognized *Bartonella* species available in the GenBank database were downloaded and used for phylogenetic analyses (Geneious software, version 10.0.6). Alignments were generated using MAFFT v.7. The final *gltA* dataset contained 55 different sequences (representative taxa) and 1199 characters (Appendix A). Phylogenetic reconstructions were conducted using Maximum likelihood (ML; PhyML v.3.0.1) and Bayesian inference (BI; MrBayes v.3.2.2) with model optimization in ModelTest v.3.06. A general time-reversible substitution model with a mixed model for among-site rate variation (GTR + Γ + I) was chosen as the best fitting sequence evolution model. Bootstrap analyses involved heuristic searches with 1000 replicates (ML). Bayesian inference analysis was run for five million generations with covarion and sampling every 100 generations. The newly read sequences were submitted to the GenBank database under the following accession numbers: MZ089835–MZ089851 and MZ089548. The maps were created in QGIS software version 3.8.3-Zanzibar.

## 5. Conclusions

In our project, we focused on three vertebrate species: the European hedgehog, the Northern white-breasted hedgehog, and the Eurasian red squirrel and their role as hosts of *Bartonella* spp. These mammals tend towards synurbization, are frequently heavily infested by ticks and fleas and are well-known as reservoir hosts for many other zoonotic pathogens, such as *Borrelia burgdorferi* s.l. Emerging vector-borne zoonotic diseases are a matter of high importance, and the cadavers of accidentally killed animals or handicapped animals that had died in rescue centers can be successfully used for pathogen monitoring. In this study, various tissues of 180 cadavers were screened by real-time PCR for the presence of *Bartonella* spp. DNA. The results were verified by multiple conventional PCR reactions, and this approach revealed a higher *Bartonella* spp. infection rate and level of diversity, which would have been missed if only a sole PCR assay targeting a single locus had been used. The highest *Bartonella* DNA infection rate was observed for *S. vulgaris* (76%), followed by *E. roumanicus* (42%) and *E. europaeus* (24%). The current study extends the known host range of *Bartonella* species with ascertained zoonotic potential, *B. washoensis* and *B. melophagi*. Moreover, two probably new, previously uncharacterized *Bartonella* species were found in hedgehogs and squirrels. We conclude that hedgehogs and squirrels are suitable hosts for various zoonotic *Bartonella* spp. and are most likely involved in their maintenance in the environment.

## Figures and Tables

**Figure 1 pathogens-10-00686-f001:**
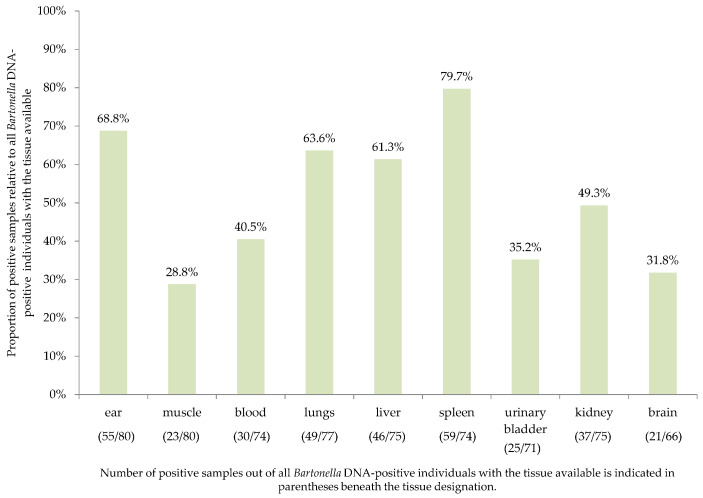
The overall infection rate of *Bartonella* spp. assessed by real-time PCR in different tissues obtained from cadavers of Eurasian red squirrels, European and Northern white-breasted hedgehogs presented as a proportion of positive samples relative to all *Bartonella* DNA-positive individuals with the tissues available.

**Figure 2 pathogens-10-00686-f002:**
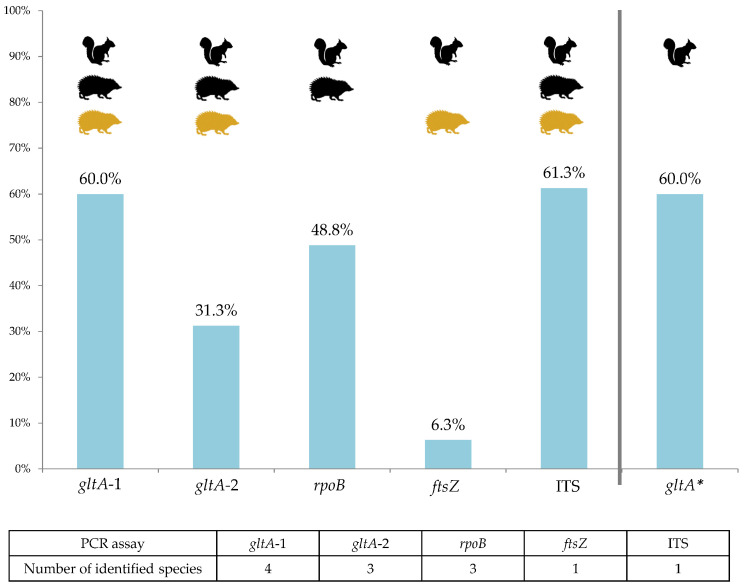
Conventional PCR sensitivity presented as a proportion of positive samples relative to all *Bartonella*-positive samples identified by six PCR assays altogether. Successful detection (including low-quality sequences) is indicated by pictograms of host species (hedgehogs: black = *E. europaeus*, yellow = *E. roumanicus*). In the case of species-specific *gltA** PCR, the proportion is calculated from *B. washoensis*-positive samples only. The number of identified *Bartonella* species for a particular PCR assay (from the total of six detected species) is summarized below the chart.

**Figure 3 pathogens-10-00686-f003:**
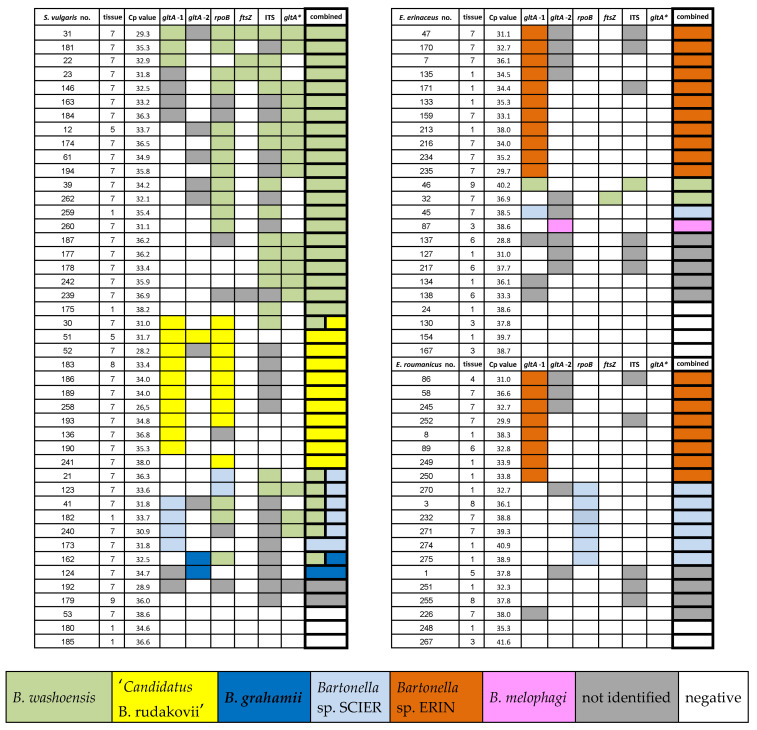
*Bartonella* species determination in real-time PCR *Bartonella*-positive samples verified by six different conventional PCR reactions and subsequent sequencing. Mixed infections = two colors in the column. Tissue: 1 = ear; 3 = muscle; 4 = blood; 5 = lungs; 6 = liver; 7 = spleen; 8 = urinary bladder; 9 = kidney. * *Bartonella washoensis-specific* PCR.

**Figure 4 pathogens-10-00686-f004:**
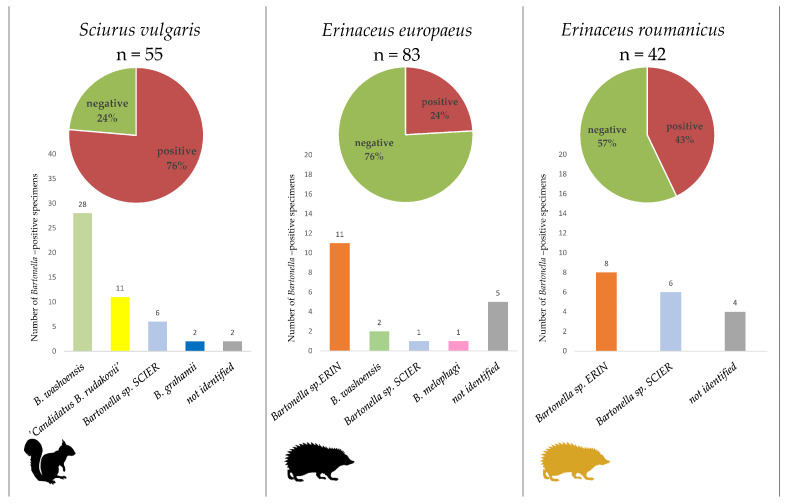
Overall infection rate of *Bartonella* spp. (pie charts) in the three tested host species as assessed by real-time PCR and verified by a set of conventional PCRs. The proportion of six detected *Bartonella* species (bar charts) based on sequencing data (summed for single and multiple infections).

**Figure 5 pathogens-10-00686-f005:**
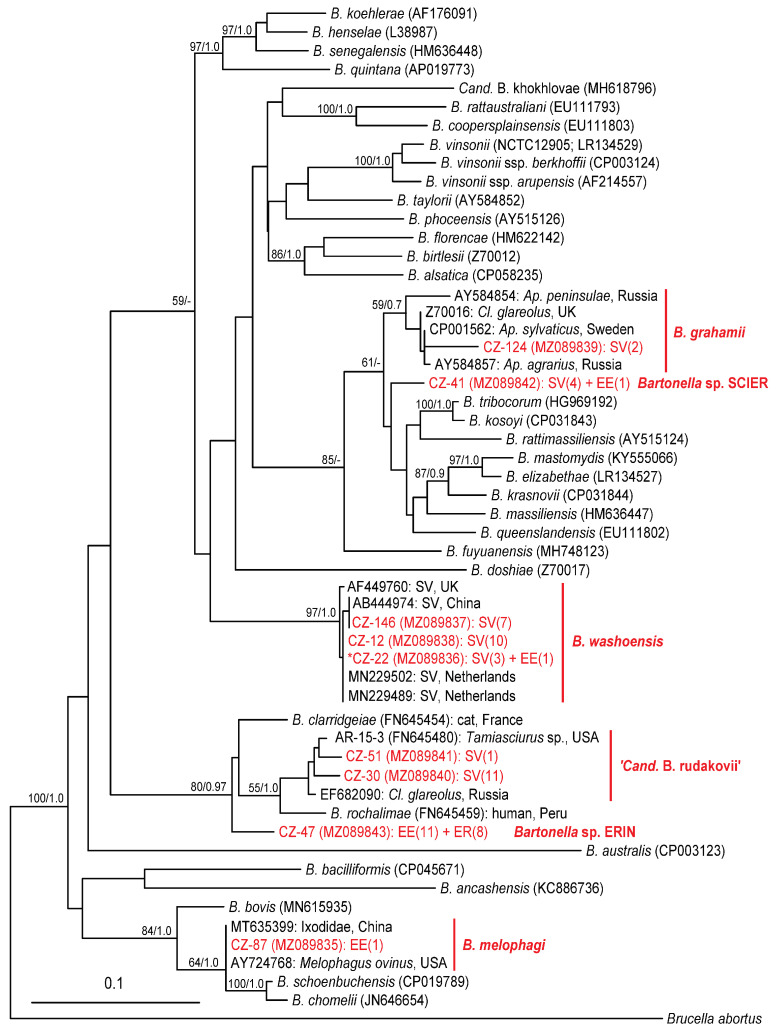
The phylogenetic tree of the *Bartonella* species based on the citrate synthase (*gltA*) gene sequences and reconstructed using the Maximum likelihood method. Statistical support at the nodes is presented as bootstrap values for Maximum likelihood/Bayesian posterior probabilities; the scale bar denotes the number of substitutions per site; the tree was rooted with the *Brucella abortus* sequence. Newly obtained sequences are shown in red; each genospecies/genotype is represented by a single sequence (incl. GenBank Accession Number). The number of positive animals with the identical sequence is presented in parentheses using the host abbreviations SV (Eurasian red squirrel, *Sciurus vulgaris*), EE (European hedgehog, *Erinaceus europaeus*) and ER (Northern white-breasted hedgehog, *E. roumanicus*). * CZ-22 is a product of PCR targeting other parts of the citrate synthase gene than species-specific PCR products CZ-12 and CZ-146. Thus, we were not able to confirm whether CZ-22 belongs to one of the genotypes represented by CZ-12 and CZ-146 or not.

**Table 1 pathogens-10-00686-t001:** Conventional PCR reactions used for *Bartonella* spp. detection and species identification.

Locus	PCR Type	Primer Name	Sequence 5′–3′	References	Product Size (bp)
*gltA*	PCR *gltA*-1	443f	GCTATGTCTGCATTCTATCA	[81]	750
		1210R	GATCYTCAATCATTTCTTTCCA	[82]	
	nested PCR *gltA*-1	443f	GCTATGTCTGCATTCTATCA	[81]	340
		781R	CCACCATGAGCTGGTCCCC	[83]	
	PCR *gltA*-2	Bhcs.781p fwd	GGGGACCAGCTCATGGTGG	[84]	370
		Bhcs.1137n rev	AATGCAAAAAGAACAGTAAACA		
	PCR *	BwgltAf	AATCAATCCAGTGCTTACTCG	[31]	625
		BwgltAr	CTGCATAGCCTGTATAGAGTT		
*rpoB*	PCR	1400F	CGCATTGGCTTACTTCGTATG	[85]	800
		2300R	GTAGACTGATTAGAACGCTG		
	nested PCR	1596F	CGCATTATGGTCGTATTTGTCC	[86]	400
		2300R	GTAGACTGATTAGAACGCTG	[85]	
*ftsZ*	PCR **	F	CATATGGTTTTCATTACTGCYGGTATGG	[86]	515
		R	TTCTTCGCGAATACGATTAGCAGCTTC		
ITS	PCR	H56s	GGGGAACCTGTGGCTGGATCAC	[87]	900–1100
		983as	TGTTCTYACAACAATGATGATG		
	nested PCR	H493as	TGAACCTCCGACCTCACGCTTATC	[87]	200–300
		321s	AGATGATGATCCCAAGCCTTCTGG		

* *Bartonella washoensis* specific PCR; ** Used in two steps with same primers and protocol.

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
