# Peer review of "Hedgehogs and Squirrels as Hosts of Zoonotic Bartonella Species"

_pathogens, 2021, doi:10.3390/pathogens10060686_

Round 1
Reviewer 1 Report
The manuscript „ Hedgehogs and squirrels as host of zoonotic Bartonella species" by Majerová et al. is interesting study from Czech Republic showing data on the distribution of Bartonella spp. in hedgehogs and squirrels in Czech Republic. I found no errors in the manuscript that require correction.
Author Response
Thank you for Your evaluation and nice words about the manuscript. As no corrections were required from Your side, no changes were done according to Your report.
Reviewer 2 Report
Authors presented an interesting study about the detection of six Bartonella genospecies in Eurasian red squirrels, European hedgehogs and Northern white-breasted hedgehogs. The publication shows a high level of applied molecular analyses, clear presentation of results and an appropriate caution in the summary part (conclusions). Moreover, structure of paper and discussion are understandable and transparent.
Congratulations!
Author Response

(The authors gave the same response as above.)

Reviewer 3 Report
The authors present data on the prevalence and diversity of Bartonella species in the cadavers of three potential hosts, Eurasian red squirrel (Sciurus vulgaris), European hedgehog (Erinaceus europaeus), and Northern white-breasted hedgehog (E. roumanicus) collected in Czechia. The authors used multiple PCR reactions targeting many loci in order to detect, identify and choose the best tissue for Bartonella spp. identification.
The study is very interesting and relevant in its field. A lot of work had been invested in the analyzes. The manuscript is well written and clear for potential readers. The study design is well described and the results are appropriately documented. I have only minor comments on the study.
Lines 128-132 should be deleted as the sentence belongs to methods and the data is repeated.
Figure 4: Interestingly, most unidentified Bartonella species were found in hedgehogs and only a few in squirrels (25% and 22% for E. europaeus and E. roumanicus, respectively vs. 4% for S. vulgaris). Do the authors have an idea why it was like this? Could you suggest an explanation?
Materials and Methods: I suggest mentioning that carcasses were collected in the Czech Republic (somewhere between lines 359-365).
Author Response
Point 1: Lines 128-132 should be deleted as the sentence belongs to methods and the data is repeated.
Response 1: We fully agree with this comment, but the reason for the repeating information is the required article structure in the Pathogens journal. Because the „Materials and methods“ part is placed after the „Results“, we feel that for readers would be really very hard to understand the text and figures without this very short mention of the used methodological approach and also without the abbreviations description (gltA, rpoB, ftsZ, ITS). We tried to use slightly different wording but we would like to keep this information in both mentioned article parts. We believe that you will understand our reasons and agree with the stated/used solution; we would really appreciate it.
Point 2: Figure 4: Interestingly, most unidentified Bartonella species were found in hedgehogs and only a few in squirrels (25% and 22% for E. europaeus and E. roumanicus, respectively vs. 4% for S. vulgaris). Do the authors have an idea why it was like this? Could you suggest an explanation?
Response 2: Thank You for this interesting question. Unfortunately, we don´t have any clear explanation for the observed differences among the host species. In the case of hedgehogs, ear tissue had the lowest Cp value and was used more frequently for conventional PCRs (30% and 39% for E. europaeus and E. roumanicus, respectively) than in squirrels (7%). Some of the tested samples, where Bartonella species was not identified, came from this tissue and it is possible, that some contamination (e.g. bacteria) from the skin surface could be also amplified by used PCR protocols and cause the mixed reading in the obtained sequences. We also cannot exclude the false-positive amplification of the host DNA itself, which was actually noticed more often in hedgehogs („false positive“ samples) than in squirrels.
However, these above-mentioned explanations are just hypothetical and we are not sure about the reasons. The total number of hosts is low (5 European hedgehogs and 4 Northern white-breasted hedgehogs vs. 2 squirrels), so we decided to not discuss this question in the manuscript.
Point 3: Materials and Methods: I suggest mentioning that carcasses were collected in the Czech Republic (somewhere between lines 359-365).
Response 3: Thank you for your suggestion. The sentence was changed on line 361 to “A total of 180 cadavers of the European hedgehog (Erinaceus europaeus), the Northern white-breasted hedgehog (Erinaceus roumanicus) and the Eurasian red squirrel (Sciurus vulgaris) were collected in various geographical regions of the Czech Republic for a project addressing the role of these species in the circulation of vector-borne pathogens and their possible use as sentinel hosts.”
Reviewer 4 Report
Line 114 : I think the "a total of 180 cadavers were collected" phrase should be put into the materials section.
From line 128 to 125 : to be modified as it is a repetition of the methods.
Figure 3 : this figure is unclear, it is grainy. I think it needs to be improved as an image.
For "Materials and methods": how did you preserve the cadavers until dissection ? How did you preserve the tissues obtained from the dissection of the corpses before the analysis ? Have you tried to carry out the isolation of the bacteria ? Please, explain why you did not perform this test.
From line 421 to line 429 : I think these lines are a repeat of the results.
The conclusions need to be broadened and improved.
Author Response
Point 1: Line 114: I think the "a total of 180 cadavers were collected" phrase should be put into the materials section.
Response 1: Thank you for Your suggestion. The phrase "a total of 180 cadavers were collected" was deleted on line 114 and the following sentence was changed accordingly. The phrase was added to the first sentence of Materials and Methods (line 359): “A total of 180 cadavers of the European hedgehog (Erinaceus europaeus), the Northern white-breasted hedgehog (Erinaceus roumanicus) and the Eurasian red squirrel (Sciurus vulgaris) were collected in various geographical regions of the Czech Republic for a project addressing the role of these species in the circulation of vector-borne pathogens and their possible use as sentinel hosts.”
Point 2: From line 128 to 125: to be modified as it is a repetition of the methods.
Response 2: We fully agree with this comment, but the reason for the repeating information is the required article structure in the Pathogens journal. Because the „Materials and methods“ part is placed after the „Results“, we feel that for readers would be really very hard to understand the text and figures without this very short mention of the used methodological approach and also without the abbreviations description (gltA, rpoB, ftsZ, ITS). We tried to use slightly different wording but we would like to keep this information in both mentioned article parts. We believe that you will understand our reasons and agree with the stated/used solution; we would really appreciate it.
Point 3: Figure 3: this figure is unclear, it is grainy. I think it needs to be improved as an image.
Response 3: Thank You for Your attentive reading of the manuscript! You are absolutely right, the image (Figure 3) was grainy and I haven´t noticed it before. It was replaced by an image of better quality.
Point 4: For "Materials and methods": how did you preserve the cadavers until dissection? How did you preserve the tissues obtained from the dissection of the corpses before the analysis? Have you tried to carry out the isolation of the bacteria? Please, explain why you did not perform this test.
Response 4: Thank You for the question. In the manuscript, we refer to two of our previous articles (Majerová et al., 2020; Lesiczka et al., 2021) related to the project, where the methodology is described in detail. Cadavers were kept at −20°C until processing. RLT buffer (1 ml; Qiagen) was added to each Eppendorf tube with a blood or tissue sample and stored at −70°C until further analysis (because RNA was also isolated for Flaviviruses detection). Tissue samples were homogenized in RLT buffer (Qiagen) containing beta-mercaptoethanol (10 μl of 14.3M betamercaptoethanol per 1 ml of RLT buffer) using sterile stainless steel beads (Qiagen) and Tissue Lyzer II. Briefly, samples of ear, skin, muscle, lung, liver, spleen, urinary bladder, kidney, and brain were prepared as 30% (w/v) suspensions. The homogenates aliquots were stored at −20°C until DNA extraction.
The phrase "Cadavers were kept at −20°C until processing“ was added to the first paragraph of "Materials and methods".
We have not tried bacteria isolation. The cadavers were sampled within a larger project (and we acknowledge it in the manuscript also in the first paragraph of "Materials and methods") addressing the role of the chosen vertebrate species in the circulation of vector-borne pathogens and their possible use as sentinels in an urban environment. Thus, our original goal was to perform a multiple PCR screening, but not the isolation of different pathogens, which would be really difficult. Moreover, the condition of most of the cadavers would not enable us to perform bacteria cultivation anyway, although we are well aware this method would be very useful for Bartonella species identification and characterization.
Point 5: From line 421 to line 429: I think these lines are a repeat of the results.
The conclusions need to be broadened and improved.
Response 5: Thank You for your suggestion. We have broadened the “Conclusions” part and removed some information that repeated the “Results” part. The current version of the “Conclusions” is:
In our project, we focused on three vertebrate species: the European hedgehog, the Northern white-breasted hedgehog, and the Eurasian red squirrel, and their role as hosts of Bartonella spp. These mammals tend towards synurbization, are frequently heavily infested by ticks and fleas, and are well-known as reservoir hosts for many other zoonotic pathogens, such as Borrelia burgdorferi s.l. Emerging vector-borne zoonotic diseases are a matter of high importance and the cadavers of accidentally killed animals or handicapped animals that had died in rescue centers can be successfully used for pathogen monitoring. In this study, various tissues of 180 cadavers were screened by real-time PCR for the presence of Bartonella spp. DNA. The results were verified by multiple conventional PCR reactions and this approach revealed a higher Bartonella spp. infection rate and level of diversity, which would have been missed if only a sole PCR assay targeting a single locus had been used. The highest Bartonella DNA infection rate was observed for S. vulgaris (76%), followed by E. roumanicus (42%) and E. europaeus (24%). The current study extends the known host range of Bartonella species with ascertained zoonotic potential, B. washoensis and B. melophagi. Moreover, two probably new, previously uncharacterized Bartonella species were found in hedgehogs and squirrels. We conclude that hedgehogs and squirrels are suitable hosts for various zoonotic Bartonella spp. and are most likely involved in their maintenance in the environment.